# ACCELERATION IN HYPERBOLIC AND SPHERICAL SPACES

## ABSTRACT

We further research on the acceleration phenomenon on Riemannian manifolds by introducing the first global first-order method that achieves the same rates as accelerated gradient descent in the Euclidean space for the optimization of smooth and geodesically convex (g-convex) or strongly g-convex functions defined on the hyperbolic space or a subset of the sphere, up to constants and log factors. To the best of our knowledge, this is the first method that is proved to achieve these rates globally on functions defined on a Riemannian manifold $\mathcal{M}$ other than the Euclidean space. Additionally, for any Riemannian manifold of bounded sectional curvature, we provide reductions from optimization methods for smooth and g-convex functions to methods for smooth and strongly g-convex functions and vice versa.

## 1 INTRODUCTION

Acceleration in convex optimization is a phenomenon that has drawn lots of attention and has yielded many important results, since the renowned Accelerated Gradient Descent (AGD) method of Nesterov (1983). Having been proved successful for deep learning Sutskever et al. (2013), among other fields, there have been recent efforts to better understand this phenomenon Allen Zhu & Orecchia (2017); Diakonikolas & Orecchia (2019); Su et al. (2016); Wibisono et al. (2016). These have yielded numerous new results going beyond convexity or the standard oracle model, in a wide variety of settings Allen-Zhu (2017; 2018a;b); Allen Zhu & Orecchia (2015); Allen Zhu et al. (2016); Allen-Zhu et al. (2017); Carmon et al. (2017); Cohen et al. (2018); Cutkosky & Sarlós (2019); Diakonikolas & Jordan (2019); Diakonikolas & Orecchia (2018); Gasnikov et al. (2019); Wang et al. (2016). This surge of research that applies tools of convex optimization to models going beyond convexity has been fruitful. One of these models is the setting of geodesically convex Riemannian optimization. In this setting, the function to optimize is geodesically convex (g-convex), i.e. convex restricted to any geodesic (cf. Definition 1.1).

Riemannian optimization, g-convex and non-g-convex alike, is an extensive area of research. In recent years there have been numerous efforts towards obtaining Riemannian optimization algorithms that share analogous properties to the more broadly studied Euclidean first-order methods: deterministic de Carvalho Bento et al. (2017); Wei et al. (2016); Zhang & Sra (2016), stochastic Hosseini & Sra (2017); Khuzani & Li (2017); Tripuraneni et al. (2018), variance-reduced Sato et al. (2017; 2019); Zhang et al. (2016), adaptive Kasai et al. (2019), saddle-point-escaping Criscitiello & Boumal (2019); Sun et al. (2019); Zhang et al. (2018); Zhou et al. (2019); Criscitiello & Boumal (2020), and projection-free methods Weber & Sra (2017; 2019), among others. Unsurprisingly, Riemannian optimization has found many applications in machine learning, including low-rank matrix completion Cambier & Absil (2016); Heidel & Schulz (2018); Mishra & Sepulchre (2014); Tan et al. (2014); Vandereycken (2013), dictionary learning Cherian & Sra (2017); Sun et al. (2017), optimization under orthogonality constraints Edelman et al. (1998), with applications to Recurrent Neural Networks Lezcano-Casado (2019); Lezcano-Casado & Martínez-Rubio (2019), robust covariance estimation in Gaussian distributions Wiesel (2012), Gaussian mixture models Hosseini & Sra (2015), operator scaling Allen-Zhu et al. (2018), and sparse principal component analysis Genicot et al. (2015); Huang & Wei (2019b); Jolliffe et al. (2003).

However, the acceleration phenomenon, largely celebrated in the Euclidean space, is still not understood in Riemannian manifolds, although there has been some progress on this topic recently (cf. Related work). This poses the following question, which is the central subject of this paper:

*Can a Riemannian first-order method enjoy the same rates as AGD in the Euclidean space?*

In this work, we provide an answer in the affirmative for functions defined on hyperbolic and spherical spaces, up to constants depending on the curvature and the initial distance to an optimum, and up to log factors. In particular, the main results of this work are the following.

**Main Results:**

- *Full acceleration.* We design algorithms that provably achieve the same rates of convergence as AGD in the Euclidean space, up to constants and log factors. More precisely, we obtain the rates $\widetilde{O}(L/\sqrt{\varepsilon})$ and $O^*(\sqrt{L/\mu}\log(\mu/\varepsilon))$ when optimizing $L$-smooth functions that are, respectively, g-convex and $\mu$-strongly g-convex, defined on the hyperbolic space or a subset of the sphere. The notation $\widetilde{O}(\cdot)$ and $O^*(\cdot)$ omits $\log(L/\varepsilon)$ and $\log(L/\mu)$ factors, respectively, and constants. Previous approaches only showed local results Zhang & Sra (2018) or obtained results with rates in between the ones obtainable by Riemannian Gradient Descent (RGD) and AGD Ahn & Sra (2020). Moreover, these previous works only apply to functions that are smooth and strongly g-convex and not to smooth functions that are only g-convex. As a proxy, we design an accelerated algorithm under a condition between of convexity and *quasar-convexity* in the constrained setting, which is of independent interest.

- *Reductions.* We present two reductions for any Riemannian manifold of bounded sectional curvature. Given an optimization method for smooth and g-convex functions they provide a method for optimizing smooth and strongly g-convex functions, and vice versa. This allows to focus on designing methods for one set of assumptions only.

It is often the case that methods and key geometric inequalities that apply to manifolds with bounded sectional curvatures are obtained from the ones existing for the spaces of constant extremal sectional curvature Grove et al. (1997); Zhang & Sra (2016; 2018). Consequently, our contribution is relevant not only because we establish an algorithm achieving global acceleration on functions defined on a manifold other than the Euclidean space, but also because understanding the constant sectional curvature case is an important step towards understanding the more general case of obtaining algorithms that optimize g-convex functions, strongly or not, defined on manifolds of bounded sectional curvature.

Our main technique for designing the accelerated method consists of mapping the function domain to a subset $\mathcal{B}$ of the Euclidean space via a geodesic map: a transformation that maps geodesics to geodesics. Given the gradient of a point $x \in \mathcal{M}$, which defines a lower bound on the function that is linear over the tangent space of $x$, we find a lower bound of the function that is linear over $\mathcal{B}$, despite the map being non-conformal, deforming distances, and breaking convexity. This allows to aggregate the lower bounds easily. We believe that effective lower bound aggregation is key to achieving Riemannian acceleration and optimality. Using this strategy, we are able to provide an algorithm along the lines of the one in Diakonikolas & Orecchia (2018) to define a continuous method that we discretize using an approximate implementation of the implicit Euler method, obtaining a method achieving the same rates as the Euclidean AGD, up to constants and log factors. Our reductions take into account the deformations produced by the geometry to generalize existing Euclidean reductions Allen Zhu & Hazan (2016); Allen Zhu & Orecchia (2017).

**Basic Geometric Definitions.** We recall basic definitions of Riemannian geometry that we use in this work. For a thorough introduction we refer to Petersen et al. (2006). A Riemannian manifold $(\mathcal{M}, \mathfrak{g})$ is a real smooth manifold $\mathcal{M}$ equipped with a metric $\mathfrak{g}$, which is a smoothly varying inner product. For $x \in \mathcal{M}$ and any two vectors $v, w \in T_x\mathcal{M}$ in the tangent space of $\mathcal{M}$, the inner product $\langle v, w \rangle_x$ is $\mathfrak{g}(v, w)$. For $v \in T_x\mathcal{M}$, the norm is defined as usual $\|v\|_x \stackrel{\text{def}}{=} \sqrt{\langle v, v \rangle_x}$. Typically, $x$ is known given $v$ or $w$, so we will just write $\langle v, w \rangle$ or $\|v\|$ if $x$ is clear from context. A geodesic is a curve $\gamma : [0, 1] \to \mathcal{M}$ of unit speed that is locally distance minimizing. A uniquely geodesic space is a space such that for every two points there is one and only one geodesic that joins them. In such a case the exponential map $\text{Exp}_x : T_x\mathcal{M} \to \mathcal{M}$ and inverse exponential map $\text{Exp}_x^{-1} : \mathcal{M} \to T_x\mathcal{M}$ are well defined for every pair of points, and are as follows. Given $x, y \in \mathcal{M}$, $v \in T_x\mathcal{M}$, and a

geodesic $\gamma$ of length $\|v\|$ such that $\gamma(0) = x$, $\gamma(1) = y$, $\gamma'(0) = v/\|v\|$, we have that $\mathrm{Exp}_x(v) = y$ and $\mathrm{Exp}_x^{-1}(y) = v$. Note, however, that $\mathrm{Exp}_x(\cdot)$ might not be defined for each $v \in T_x\mathcal{M}$. We denote by $d(x, y)$ the distance between $x$ and $y$. Its value is the same as $\|\mathrm{Exp}_x^{-1}(y)\|$. Given a 2-dimensional subspace $V \subseteq T_x\mathcal{M}$, the sectional curvature at $x$ with respect to $V$ is defined as the Gauss curvature of the manifold $\mathrm{Exp}_x(V)$ at $x$.

**Notation.** Let $\mathcal{M}$ be a manifold and let $\mathcal{B} \subseteq \mathbb{R}^d$. We denote by $h : \mathcal{M} \to \mathcal{B}$ a geodesic map Kreyszig (1991), which is a diffeomorphism such that the image and the inverse image of a geodesic is a geodesic. Usually, given an initial point $x_0$ of our algorithm, we will have $h(x_0) = 0$. Given a point $x \in \mathcal{M}$ we use the notation $\tilde{x} = h(x)$ and vice versa, any point in $\mathcal{B}$ will use a tilde. Given two points $x, y \in \mathcal{M}$ and a vector $v \in T_x\mathcal{M}$ in the tangent space of $x$, we use the formal notation $\langle v, y - x \rangle \overset{\mathrm{def}}{=} -\langle v, x - y \rangle \overset{\mathrm{def}}{=} \langle v, \mathrm{Exp}_x^{-1}(y) \rangle$. Given a vector $v \in T_x\mathcal{M}$, we call $\tilde{v} \in \mathbb{R}^d$ the vector of the same norm such that $\{\tilde{x} + \tilde{\lambda}\tilde{v} | \tilde{\lambda} \in \mathbb{R}^+, \tilde{x} + \tilde{\lambda}\tilde{v} \in \mathcal{B}\} = \{h(\mathrm{Exp}_x(\lambda v)) | \lambda \in I \subseteq \mathbb{R}^+\}$, for some interval $I$. Likewise, given $x$ and a vector $\tilde{v} \in \mathbb{R}^d$, we define $v \in T_x\mathcal{M}$. Let $x^*$ be any minimizer of $F : \mathcal{M} \to \mathbb{R}$. We denote by $R \geq d(x_0, x^*)$ a bound on the distance between $x^*$ and the initial point $x_0$. Note that this implies that $x^* \in \mathrm{Exp}_{x_0}(\bar{B}(0, R))$, for the closed ball $\bar{B}(0, R) \subseteq T_{x_0}\mathcal{M}$. Consequently, we will work with the manifold that is a subset of a $d$-dimensional complete and simply connected manifold of constant sectional curvature $K$, namely a subset of the hyperbolic space or sphere Petersen et al. (2006), defined as $\mathrm{Exp}_{x_0}(\bar{B}(0, R))$, with the inherited metric. Denote by $\mathcal{H}$ this manifold in the former case and $\mathcal{S}$ in the latter, and note that we are not making explicit the dependence on $d$, $R$ and $K$. We want to work with the standard choice of uniquely geodesic manifolds Ahn & Sra (2020); Liu et al. (2017); Zhang & Sra (2016; 2018). Therefore, in the case that the manifold is $\mathcal{S}$, we restrict ourselves to $R < \pi/2\sqrt{K}$, so $\mathcal{S}$ is contained in an open hemisphere. The big $O$ notations $\widetilde{O}(\cdot)$ and $O^*(\cdot)$ omit $\log(L/\varepsilon)$ and $\log(L/\mu)$ factors, respectively, and constant factors depending on $R$ and $K$.

We define now the main properties that will be assumed on the function $F$ to be minimized.

**Definition 1.1 (Geodesic Convexity and Smoothness).** Let $F : \mathcal{M} \to \mathbb{R}$ be a differentiable function defined on a Riemannian manifold $(\mathcal{M}, \mathfrak{g})$. Given $L \geq \mu > 0$, we say that $F$ is $L$-smooth, and respectively $\mu$-strongly g-convex, if for any two points $x, y \in \mathcal{M}$, $F$ satisfies

$$F(y) \leq F(x) + \langle \nabla F(x), y - x \rangle + \frac{L}{2}d(x, y)^2, \text{ resp. } F(y) \geq F(x) + \langle \nabla F(x), y - x \rangle + \frac{\mu}{2}d(x, y)^2.$$

We say $F$ is g-convex if the second inequality above, i.e. $\mu$-strong g-convexity, is satisfied with $\mu = 0$. Note that we have used the formal notation above for the subtraction of points in the inner product.

**Comparison with Related Work.** There are a number of works that study the problem of first-order acceleration in Riemannian manifolds of bounded sectional curvature. The first study is Liu et al. (2017). In this work, the authors develop an accelerated method with the same rates as AGD for both g-convex and strongly g-convex functions, provided that at each step a given nonlinear equation can be solved. No algorithm for solving this equation has been found and, in principle, it could be intractable or infeasible. In Alimisis et al. (2019) a continuous method analogous to the continuous approach to accelerated methods is presented, but it is not known if there exists an accelerated discretization of it. In Alimisis et al. (2020), an algorithm presented is claimed to enjoy an accelerated rate of convergence, but fails to provide convergence when the function value gets below a potentially large constant that depends on the manifold and smoothness constant. In Huang & Wei (2019a) an accelerated algorithm is presented but relying on strong geometric inequalities that are not proved to be satisfied. Zhang & Sra (2018) obtain a *local* algorithm that optimizes $L$-smooth and $\mu$-strongly g-convex functions achieving the same rates as AGD in the Euclidean space, up to constants. That is, the initial point needs to start close to the optimum, $O((\mu/L)^{3/4})$ close, to be precise. Their approach consists of adapting Nesterov's estimate sequence technique by keeping a quadratic on $T_{x_t}\mathcal{M}$ that induces on $\mathcal{M}$ a regularized lower bound on $F(x^*)$ via $\mathrm{Exp}_{x_t}(\cdot)$. They aggregate the information yielded by the gradient to it, and use a geometric lemma to find a quadratic in $T_{x_{t+1}}\mathcal{M}$ whose induced function lower bounds the other one. Ahn & Sra (2020) generalize the previous algorithm and, by using similar ideas for the lower bound, they adapt it to work globally, obtaining strictly better rates than RGD, recovering the local acceleration of the previous paper, but not achieving global rates comparable to the ones of AGD. In fact, they prove that their algorithm eventually decreases the function value at a rate close to AGD but this can take as many iterations as the ones needed by RGD to minimize the function. In our work, we take a step back and focus

on the constant sectional curvature case to provide a global algorithm that achieves the same rates as AGD, up to constants and log factors. It is common to characterize the properties of spaces of bounded sectional curvature by using the ones of the spaces of constant extremal sectional curvature Grove et al. (1997); Zhang & Sra (2016; 2018), which makes the study of the constant sectional curvature case critical to the development of full accelerated algorithms in the general bounded sectional curvature case. Additionally, our work studies g-convexity besides strong g-convexity.

Another related work is the *approximate duality gap technique* Diakonikolas & Orecchia (2019), which presents a unified view of the analysis of first-order methods for the optimization of convex functions defined in the Euclidean space. It defines a continuous duality gap and by enforcing a natural invariant, it obtains accelerated continuous dynamics and their discretizations for most classical first-order methods. A derived work Diakonikolas & Orecchia (2018) obtains acceleration in a fundamentally different way from previous acceleration approaches, namely using an approximate implicit Euler method for the discretization of the acceleration dynamics. The convergence analysis of Theorem 2.4 is inspired by these two works. We will see in the sequel that, for our manifolds of interest, g-convexity is related to a model known in the literature as quasar-convexity or weak-quasi-convexity Guminov & Gasnikov (2017); Hinder et al. (2019); Nesterov et al. (2018).

## 2 ALGORITHM

We study the minimization problem $\min_{x \in \mathcal{M}} F(x)$ with a gradient oracle, for a smooth function $F : \mathcal{M} \to \mathbb{R}$ that is g-convex or strongly g-convex. In this section, $\mathcal{M}$ refers to a manifold that can be $\mathcal{H}$ or $\mathcal{S}$, i.e. the subset of the hyperbolic space or sphere $\mathrm{Exp}_{x_0}(\bar{B}(0, R))$, for an initial point $x_0$. For simplicity, we do not use subdifferentials so we assume $F : \mathcal{M} \to \mathbb{R}$ is a differentiable function that is defined over the manifold of constant sectional curvature $\mathcal{M}' \overset{\text{def}}{=} \mathrm{Exp}_{x_0}(B(0, R'))$, for an $R' > R$, and we avoid writing $F : \mathcal{M}' \to \mathbb{R}$. We defer the proofs of the lemmas and theorems in this and following sections to the supplementary material. We assume without loss of generality that the sectional curvature of $\mathcal{M}$ is $K \in \{1, -1\}$, since for any other value of $K$ and any function $F : \mathcal{M} \to \mathbb{R}$ defined on such a manifold, we can reparametrize $F$ by a rescaling, so it is defined over a manifold of constant sectional curvature $K \in \{1, -1\}$. The parameters $L$, $\mu$ and $R$ are rescaled accordingly as a function of $K$, cf. Remark C.1. We denote the special cosine by $C_K(\cdot)$, which is $\cos(\cdot)$ if $K = 1$ and $\cosh(\cdot)$ if $K = -1$. We define $\mathcal{X} = h(\mathcal{M}) \subseteq \mathcal{B} \subseteq \mathbb{R}^d$. We use classical geodesic maps for the manifolds that we consider: the Gnomonic projection for $\mathcal{S}$ and the Beltrami-Klein projection for $\mathcal{H}$ Greenberg (1993). They map an open hemisphere and the hyperbolic space of curvature $K \in \{1, -1\}$ to $\mathcal{B} = \mathbb{R}^d$ and $\mathcal{B} = B(0, 1) \subseteq \mathbb{R}^d$, respectively. We will derive our results from the following characterization Greenberg (1993). Let $\tilde{x}, \tilde{y} \in \mathcal{B}$ be two points. Recall that we denote $x = h^{-1}(\tilde{x}), y = h^{-1}(\tilde{y}) \in \mathcal{M}$. Then we have that $d(x, y)$, the distance between $x$ and $y$ with the metric of $\mathcal{M}$, satisfies

$$C_K(d(x, y)) = \frac{1 + K\langle \tilde{x}, \tilde{y} \rangle}{\sqrt{1 + K\|\tilde{x}\|^2} \cdot \sqrt{1 + K\|\tilde{y}\|^2}}. \tag{1}$$

Observe that the expression is symmetric with respect to rotations. In particular, the symmetry implies $\mathcal{X}$ is a closed ball of radius $\tilde{R}$, with $C_K(R) = (1 + K\tilde{R}^2)^{-1/2}$.

Consider a point $x \in \mathcal{M}$ and the lower bound provided by the g-convexity assumption when computing $\nabla F(x)$. Dropping the $\mu$ term in case of strong g-convexity, this bound is linear over $T_x \mathcal{M}$. We would like our algorithm to aggregate effectively the lower bounds it computes during the course of the optimization. The deformations of the geometry make it a difficult task, despite the fact that we have a simple description of each individual lower bound. We deal with this problem in the following way: our approach is to obtain a lower bound that is looser by a constant depending on $R$, and that is linear over $\mathcal{B}$. In this way the aggregation becomes easier. Then, we are able to combine this lower bound with decreasing upper bounds in the fashion some other accelerated methods work in the Euclidean space Allen Zhu & Orecchia (2017); Diakonikolas & Orecchia (2018; 2019); Nesterov (1983). Alternatively, we can see the approach in this work as the constrained non-convex optimization problem of minimizing the function $f : \mathcal{X} \to \mathbb{R}$, $\tilde{x} \mapsto F(h^{-1}(\tilde{x}))$:

$$\text{minimize } f(\tilde{x}), \quad \text{for } \tilde{x} \in \mathcal{X}.$$

In the rest of the section, we will focus on the g-convex case. For simplicity, instead of solving the strongly g-convex case directly in an analogous way by finding a lower bound that is quadratic over $\mathcal{B}$, we rely on the reductions of Section 3 to obtain the accelerated algorithm in this case.

The following two lemmas show that finding the aforementioned linear lower bound is possible, and is defined as a function of $\nabla f(\tilde{x})$. We first gauge the deformations caused by the geodesic map $h$. Distances are deformed, the map $h$ is not conformal and, in spite of it being a geodesic map, the image of the geodesic $\mathrm{Exp}_x(\lambda \nabla F(x))$ is not mapped into the image of the geodesic $\tilde{x} + \tilde{\lambda} \nabla f(\tilde{x})$, i.e. the direction of the gradient changes. We are able to find the linear lower bound after bounding these deformations.

**Lemma 2.1.** *Let $x, y \in \mathcal{M}$ be two different points, and in part b) different from $x_0$. Let $\tilde{\alpha}$ be the angle $\angle \tilde{x}_0 \tilde{x} \tilde{y}$, formed by the vectors $\tilde{x}_0 - \tilde{x}$ and $\tilde{y} - \tilde{x}$. Let $\alpha$ be the corresponding angle between the vectors $\mathrm{Exp}_x^{-1}(x_0)$ and $\mathrm{Exp}_x^{-1}(y)$. Assume without loss of generality that $\tilde{x} \in \mathrm{span}\{\tilde{e}_1\}$ and $\nabla f(\tilde{x}) \in \mathrm{span}\{\tilde{e}_1, \tilde{e}_2\}$ for the canonical orthonormal basis $\{\tilde{e}_i\}_{i=1}^d$. Let $e_i \in T_x\mathcal{M}$ be the unit vector such that $h$ maps the image of the geodesic $\mathrm{Exp}_x(\lambda e_i)$ to the image of the geodesic $\tilde{x} + \tilde{\lambda} e_i$, for $i = 1, \ldots, d$, and $\lambda, \tilde{\lambda} \geq 0$. Then, the following holds.*

a) *Distance deformation:*

$$KC_K^2(R) \leq K \frac{d(x,y)}{\|\tilde{x} - \tilde{y}\|} \leq K.$$

b) *Angle deformation:*

$$\sin(\alpha) = \sin(\tilde{\alpha})\sqrt{\frac{1 + K\|\tilde{x}\|^2}{1 + K\|\tilde{x}\|^2 \sin^2(\tilde{\alpha})}}, \qquad \cos(\alpha) = \cos(\tilde{\alpha})\sqrt{\frac{1}{1 + K\|\tilde{x}\|^2 \sin^2(\tilde{\alpha})}}.$$

c) *Gradient deformation:*

$$\nabla F(x) = (1 + K\|\tilde{x}\|^2)\nabla f(\tilde{x})_1 e_1 + \sqrt{1 + K\|\tilde{x}\|^2}\nabla f(\tilde{x})_2 e_2 \quad and \quad e_i \perp e_j \text{ for } i \neq j.$$

*And if $v \in T_x\mathcal{M}$ is a vector normal to $\nabla F(x)$, then $\tilde{v}$ is normal to $\nabla f(x)$.*

The following uses the deformations described in the previous lemma to obtain the linear lower bound on the function, given a gradient at a point $\tilde{x}$. Note that Lemma 2.1.c implies that we have $\langle \nabla f(\tilde{x}), \tilde{y} - \tilde{x} \rangle = 0$ if and only if $\langle \nabla F(x), y - x \rangle = 0$. In the proof we lower bound, generally, linear functions defined on $T_x\mathcal{M}$ by linear functions in the Euclidean space $\mathcal{B}$. This generality allows to obtain a result with constants that only depends on $R$.

**Lemma 2.2.** *Let $F : \mathcal{M} \to \mathbb{R}$ be a differentiable function and let $f = F \circ h^{-1}$. Then, there are constants $\gamma_{\mathrm{n}}, \gamma_{\mathrm{p}} \in (0, 1]$ depending on $R$ such that for all $x, y \in \mathcal{M}$ satisfying $\langle \nabla f(\tilde{x}), \tilde{y} - \tilde{x} \rangle \neq 0$ we have:*

$$\gamma_{\mathrm{p}} \leq \frac{\langle \nabla F(x), y - x \rangle}{\langle \nabla f(\tilde{x}), \tilde{y} - \tilde{x} \rangle} \leq \frac{1}{\gamma_{\mathrm{n}}}. \tag{2}$$

*In particular, if $F$ is g-convex we have:*

$$
\begin{aligned}
f(\tilde{x}) + \frac{1}{\gamma_{\mathrm{n}}}\langle \nabla f(\tilde{x}), \tilde{y} - \tilde{x} \rangle \leq f(\tilde{y}) \quad & \text{if } \langle \nabla f(\tilde{x}), \tilde{y} - \tilde{x} \rangle \leq 0, \\
f(\tilde{x}) + \gamma_{\mathrm{p}}\langle \nabla f(\tilde{x}), \tilde{y} - \tilde{x} \rangle \leq f(\tilde{y}) \quad & \text{if } \langle \nabla f(\tilde{x}), \tilde{y} - \tilde{x} \rangle \geq 0.
\end{aligned}
\tag{3}
$$

The two inequalities in (3) show the linear lower bound. Only the first one is needed to bound $f(\tilde{x}^*) = F(x^*)$. The first inequality applied to $\tilde{y} = \tilde{x}^*$ defines a model known in the literature as quasar-convexity or weak-quasi-convexity Guminov & Gasnikov (2017); Hinder et al. (2019); Nesterov et al. (2018), for which accelerated algorithms exist in the *unconstrained case*, provided smoothness is also satisfied. However, to the best of our knowledge, there is no known algorithm for solving the constrained case in an accelerated way. The condition in (3) is, trivially, a relaxation of convexity that is stronger than quasar-convexity. We will make use of (3) in order to obtain acceleration in the constrained setting. This is of independent interest. Recall that we need the constraint to guarantee bounded deformation due to the geometry. We also require smoothness of $f$. The following lemma shows that $f$ is as smooth as $F$ up to a constant depending on $R$.

**Lemma 2.3.** *Let $F : \mathcal{M} \to \mathbb{R}$ be an L-smooth function and $f = F \circ h^{-1}$. Assume there is a point $x^* \in \mathcal{M}$ such that $\nabla F(x^*) = 0$. Then $f$ is $O(L)$-smooth.*

Using the *approximate duality gap technique* Diakonikolas & Orecchia (2019) we obtain accelerated continuous dynamics, for the optimization of the function $f$. Then we adapt AXGD to obtain an accelerated discretization. AXGD Diakonikolas & Orecchia (2018) is a method that is based on implicit Euler discretization of continuous accelerated dynamics and is fundamentally different from AGD and techniques as Linear Coupling Allen Zhu & Orecchia (2017) or Nesterov's estimate sequence Nesterov (1983). The latter techniques use a balancing gradient step at each iteration and our use of a looser lower bound complicates guaranteeing keeping the gradient step within the constraints. We state the accelerated theorem and provide a sketch of the proof in Section 2.1.

**Theorem 2.4.** *Let $Q \subseteq \mathbb{R}^d$ be a convex set of diameter $2R$. Let $f : Q \to \mathbb{R}$ be an $\tilde{L}$-smooth function satisfying (3) with constants $\gamma_n, \gamma_p \in (0, 1]$. Assume there is a point $\tilde{x}^* \in Q$ such that $\nabla f(\tilde{x}^*) = 0$.*

*Then, we can obtain an $\varepsilon$-minimizer of $f$ using $\widetilde{O}(\sqrt{\tilde{L}/(\gamma_n^2 \gamma_p \varepsilon)})$ queries to the gradient oracle of $f$.*

Finally, we have Riemannian acceleration as a direct consequence of Theorem 2.4, Lemma 2.2 and Lemma 2.3.

**Theorem 2.5 (g-Convex Acceleration).** *Let $F : \mathcal{M} \to \mathbb{R}$ be an $L$-smooth and g-convex function and assume there is a point $x^* \in \mathcal{M}$ satisfying $\nabla F(x^*) = 0$. Algorithm 1 computes a point $x_t \in \mathcal{M}$ satisfying $F(x_t) - F(x^*) \leq \varepsilon$ using $\widetilde{O}(\sqrt{L/\varepsilon})$ queries to the gradient oracle.*

We observe that if there is a geodesic map mapping a manifold into a convex subset of the Euclidean space then the manifold must necessarily have constant sectional curvature, cf. Beltrami's Theorem Busemann & Phadke (1984); Kreyszig (1991). This precludes a straightforward generalization from our method to the case of non-constant bounded sectional curvature.

---

**Algorithm 1** Accelerated g-Convex Minimization

---

**Input:** Smooth and g-convex function $F : \mathcal{M} \to \mathbb{R}$, for $\mathcal{M} = \mathcal{H}$ or $\mathcal{M} = \mathcal{S}$.
    Initial point $x_0$; Constants $\tilde{L}, \gamma_p, \gamma_n$. Geodesic map $h$ satisfying (1) and $h(x_0) = 0$.
    Bound on the distance to a minimum $R \geq d(x_0, x^*)$. Accuracy $\varepsilon$ and number of iterations $t$.

1:  $\mathcal{X} \stackrel{\text{def}}{=} h(\text{Exp}_{x_0}(B(0, R))) \subseteq \mathcal{B}$;    $f \stackrel{\text{def}}{=} F \circ h^{-1}$     and     $\psi(\tilde{x}) \stackrel{\text{def}}{=} \frac{1}{2}\|\tilde{x}\|^2$
2:  $\tilde{z}_0 \leftarrow \nabla\psi(\tilde{x}_0)$;    $A_0 \leftarrow 0$
3: **for** $i$ **from** $0$ to $t - 1$ **do**
4:     $a_{i+1} \leftarrow (i+1)\gamma_n^2 \gamma_p / 2\tilde{L}$
5:     $A_{i+1} \leftarrow A_i + a_{i+1}$
6:     $\lambda \leftarrow \text{BinaryLineSearch}(\tilde{x}_i, \tilde{z}_i, f, \mathcal{X}, a_{i+1}, A_i, \varepsilon, \tilde{L}, \gamma_n, \gamma_p)$  (cf. Algorithm 2 in Appendix A)
7:     $\tilde{\chi}_i \leftarrow (1-\lambda)\tilde{x}_i + \lambda\nabla\psi^*(\tilde{z}_i)$
8:     $\tilde{\zeta}_i \leftarrow \tilde{z}_i - (a_{i+1}/\gamma_n)\nabla f(\tilde{\chi}_i)$
9:     $\tilde{x}_{i+1} \leftarrow (1-\lambda)\tilde{x}_i + \lambda\nabla\psi^*(\tilde{\zeta}_i)$            $\left[\nabla\psi^*(\tilde{p}) = \arg\min_{\tilde{z}\in\mathcal{X}}\{\|\tilde{z}-\tilde{p}\|\} = \Pi_{\mathcal{X}}(\tilde{p})\right]$
10:    $\tilde{z}_{i+1} \leftarrow \tilde{z}_i - (a_{i+1}/\gamma_n)\nabla f(\tilde{x}_{i+1})$
11: **end for**
12: return $x_t$.

---

### 2.1   SKETCH OF THE PROOF OF THEOREM 2.4.

Inspired by the *approximate duality gap technique* Diakonikolas & Orecchia (2019), let $\alpha_t$ be an increasing function of time $t$, and denote $A_t = \int_{t_0}^t d\alpha_\tau = \int_{t_0}^t \dot{\alpha}_\tau d\tau$. We define a continuous method that keeps a solution $\tilde{x}_t$, along with a differentiable upper bound $U_t$ on $f(x_t)$ and a lower bound $L_t$ on $f(\tilde{x}^*)$. In our case $f$ is differentiable so we can just take $U_t = f(x_t)$. The lower bound comes from

$$f(\tilde{x}^*) \geq \frac{\int_{t_0}^t f(\tilde{x}_\tau)d\alpha_\tau}{A_t} + \frac{\int_{t_0}^t \frac{1}{\gamma_n}\langle\nabla f(\tilde{x}_\tau), \tilde{x}^* - \tilde{x}_\tau\rangle d\alpha_\tau}{A_t}, \tag{4}$$

after applying some desirable modifications, like regularization with a 1-strongly convex function $\psi$ and removing the unknown $\tilde{x}^*$ by taking a minimum over $\mathcal{X}$. Note (4) comes from averaging (3) for $\tilde{y} = \tilde{x}^*$. Then, if we define the gap $G_t = U_t - L_t$ and design a method that forces $\alpha_t G_t$ to be non-increasing, we can deduce $f(x_t) - f(x^*) \leq G_t \leq \alpha_{t_0} G_{t_0}/\alpha_t$. By forcing $\frac{d}{dt}(\alpha_t G_t) = 0$, we naturally obtain the following continuous dynamics, where $z_t$ is a mirror point and $\psi^*$ is the Fenchel

dual of $\psi$, cf. Definition A.2.

$$\dot{\tilde{z}}_t = -\frac{1}{\gamma_{\mathrm{n}}}\dot{\alpha}_t\nabla f(\tilde{x}_t); \quad \dot{\tilde{x}}_t = \frac{1}{\gamma_{\mathrm{n}}}\dot{\alpha}_t\frac{\nabla\psi^*(\tilde{z}_t) - \tilde{x}_t}{\alpha_t}; \quad \tilde{z}_{t_0} = \nabla\psi(\tilde{x}_{t_0}), \tilde{x}_{t_0} \in \mathcal{X} \tag{5}$$

We note that except for the constant $\gamma_{\mathrm{n}}$, these dynamics match the accelerated dynamics used in the optimization of convex functions Diakonikolas & Orecchia (2019; 2018); Krichene et al. (2015). The AXGD algorithm Diakonikolas & Orecchia (2018), designed for the accelerated optimization of convex functions, discretizes the latter dynamics following an approximate implementation of implicit Euler discretization. This has the advantage of not needing a gradient step per iteration to compensate for some positive discretization error. Note that in our case we must use (3) instead of convexity for a discretization. We are able to obtain the following discretization coming from an approximate implicit Euler discretization:

$$\begin{cases} \tilde{\chi}_i = \frac{\hat{\gamma}_i A_i}{A_i\hat{\gamma}_i + a_{i+1}/\gamma_{\mathrm{n}}}\tilde{x}_i + \frac{a_{i+1}/\gamma_{\mathrm{n}}}{A_i\hat{\gamma}_i + a_{i+1}/\gamma_{\mathrm{n}}}\nabla\psi^*(\tilde{z}_i); \quad \tilde{\zeta}_i = \tilde{z}_i - \frac{a_{i+1}}{\gamma_{\mathrm{n}}}\nabla f(\tilde{\chi}_i) \\ \tilde{x}_{i+1} = \frac{\hat{\gamma}_i A_i}{A_i\hat{\gamma}_i + a_{i+1}/\gamma_{\mathrm{n}}}\tilde{x}_i + \frac{a_{i+1}/\gamma_{\mathrm{n}}}{A_i\hat{\gamma}_i + a_{i+1}/\gamma_{\mathrm{n}}}\nabla\psi^*(\tilde{\zeta}_i); \quad \tilde{z}_{i+1} = \tilde{z}_i - \frac{a_{i+1}}{\gamma_{\mathrm{n}}}\nabla f(\tilde{x}_{i+1}) \end{cases} \tag{6}$$

where $\hat{\gamma}_i \in [\gamma_{\mathrm{p}}, 1/\gamma_{\mathrm{n}}]$ is a parameter, $\tilde{x}_0 \in \mathcal{X}$ is an arbitrary point, $\tilde{z}_0 = \nabla\psi(\tilde{x}_0)$ and now $\alpha_t$ is a discrete measure and $\dot{\alpha}_t$ is a weighted sum of Dirac delta functions $\dot{\alpha}_t = \sum_{i=1}^{\infty} a_i\delta(t - (t_0 + i - 1))$. Compare (6) with the discretization in AXGD Diakonikolas & Orecchia (2018) that is equal to our discretization but with no $\gamma_{\mathrm{n}}$ or $\hat{\gamma}_i$. Or equivalently with $\hat{\gamma}_i = 1/\gamma_{\mathrm{n}}$ and with no $\gamma_{\mathrm{n}}$ for the mirror descent updates of $\tilde{\zeta}_i$ and $\tilde{z}_{i+1}$. However, not having convexity, in order to have per-iteration discretization error less than $\hat{\varepsilon}/A_T$, we require $\hat{\gamma}_i$ to be such that $\tilde{x}_{i+1}$ satisfies

$$f(\tilde{x}_{i+1}) - f(\tilde{x}_i) \le \hat{\gamma}_i\langle\nabla f(\tilde{x}_{i+1}), \tilde{x}_{i+1} - \tilde{x}_i\rangle + \hat{\varepsilon}, \tag{7}$$

where $\hat{\varepsilon}$ is chosen so that the accumulated discretization error is $< \varepsilon/2$, after having performed the steps necessary to obtain an $\varepsilon/2$ minimizer. We would like to use (3) to find such a $\hat{\gamma}_i$ but we need to take into account that we only know $\tilde{x}_{i+1}$ a posteriori. Indeed, using (3) we conclude that setting $\hat{\gamma}_i$ to $1/\gamma_{\mathrm{n}}$ or $\gamma_{\mathrm{p}}$ then we either satisfy (7) or there is a point $\hat{\gamma}_i \in (\gamma_{\mathrm{p}}, 1/\gamma_{\mathrm{n}})$ for which $\langle\nabla f(\tilde{x}_{i+1}), \tilde{x}_{i+1} - \tilde{x}_i\rangle = 0$, which satisfies the equation for $\hat{\varepsilon} = 0$. Then, using smoothness of $f$, existence of $x^*$ (that satisfies $\nabla f(x^*) = 0$), and boundedness of $\mathcal{X}$ we can guarantee that a binary search finds a point satisfying (7) in $O(\log(\tilde{L}i/\gamma_n\hat{\varepsilon}))$ iterations. Each iteration of the binary search requires to run (6), that is, one step of the discretization. Computing the final discretization error, we obtain acceleration after choosing appropriate learning rates $a_i$. Algorithm 1 contains the pseudocode of this algorithm along with the reduction of the problem from minimizing $F$ to minimizing $f$. We chose $\psi(\tilde{x}) \overset{\text{def}}{=} \frac{1}{2}\|\tilde{x}\|^2$ as our strongly convex regularizer.

## 3 REDUCTIONS

The construction of reductions proves to be very useful in order to facilitate the design of algorithms in different settings. Moreover, reductions are a helpful tool to infer new lower bounds without extra ad hoc analysis. We present two reductions. We will see in Corollary 3.2 and Example 3.4 that one can obtain full accelerated methods to minimize smooth and strongly g-convex functions from methods for smooth and g-convex functions and vice versa. These are generalizations of some reductions designed to work in the Euclidean space Allen Zhu & Hazan (2016); Allen Zhu & Orecchia (2017). The reduction to strongly g-convex functions takes into account the effect of the deformation of the space on the strong convexity of the function $F_y(x) = d(x, y)^2/2$, for $x, y \in \mathcal{M}$. The reduction to g-convexity requires the rate of the algorithm that applies to g-convex functions to be proportional to the distance between the initial point and the optimum $d(x_0, x^*)$. The proofs of the statements in this section can be found in the supplementary material. We will use $\text{Time}_{\mathrm{ns}}(\cdot)$ and $\text{Time}(\cdot)$ to denote the time algorithms $\mathcal{A}_{\mathrm{ns}}$ and $\mathcal{A}$ below require, respectively, to perform the tasks we define below.

**Theorem 3.1.** *Let $\mathcal{M}$ be a Riemannian manifold, let $F : \mathcal{M} \to \mathbb{R}$ be an $L$-smooth and $\mu$-strongly g-convex function, and let $x^*$ be its minimizer. Let $x_0$ be a starting point such that $d(x_0, x^*) \le R$. Suppose we have an algorithm $\mathcal{A}_{\mathrm{ns}}$ to minimize $F$, such that in time $T = \text{Time}_{\mathrm{ns}}(L, \mu, R)$ it produces a point $\hat{x}_T$ satisfying $F(\hat{x}_T) - F(x^*) \le \mu \cdot d(x_0, x^*)^2/4$. Then we can compute an $\varepsilon$-minimizer of $F$ in time $O(\text{Time}_{\mathrm{ns}}(L, \mu, R)\log(R^2\mu/\varepsilon))$.*

Theorem 3.1 implies that if we forget about the strong g-convexity of a function and we treat it as it is just g-convex we can run in stages an algorithm designed for optimizing g-convex functions. The

fact that the function is strongly g-convex is only used between stages, as the following corollary shows by making use of Algorithm 1.

**Corollary 3.2.** *We can compute an $\varepsilon$-minimizer of an $L$-smooth and $\mu$-strongly g-convex function $F : \mathcal{M} \to \mathbb{R}$ in $O^*(\sqrt{L/\mu}\log(\mu/\varepsilon))$ queries to the gradient oracle, where $\mathcal{M} = \mathcal{S}$ or $\mathcal{M} = \mathcal{H}$.*

We note that in the strongly convex case, by decreasing the function value by a factor we can guarantee we decrease the distance to $x^*$ by another factor, so we can periodically recenter the geodesic map to reduce the constants produced by the deformations of the geometry, see the proof of Corollary 3.2. Finally, we show the reverse reduction.

**Theorem 3.3.** *Let $\mathcal{M}$ be a Riemannian manifold of bounded sectional curvature, let $F : \mathcal{M} \to \mathbb{R}$ be an $L$-smooth and g-convex function, and assume there is a point $x^* \in \mathcal{M}$ such that $\nabla F(x^*) = 0$. Let $x_0$ be a starting point such that $d(x_0, x^*) \leq R$ and let $\Delta$ satisfy $F(x_0) - F(x^*) \leq \Delta$. Assume we have an algorithm $\mathcal{A}$ that given an $L$-smooth and $\mu$-strongly g-convex function $\hat{F} : \mathcal{M} \to \mathbb{R}$, with minimizer in $\mathrm{Exp}_{x_0}(\bar{B}(0, R))$, and any initial point $\hat{x}_0 \in \mathcal{M}$ produces a point $\hat{x} \in \mathrm{Exp}_{x_0}(\bar{B}(0, R))$ in time $\hat{T} = \mathrm{Time}(L, \mu, \mathcal{M}, R)$ satisfying $\hat{F}(\hat{x}) - \min_{x \in \mathcal{M}} \hat{F}(x) \leq (\hat{F}(\hat{x}_0) - \min_{x \in \mathcal{M}} \hat{F}(x))/4$. Let $T = \lceil \log_2(\Delta/\varepsilon)/2 \rceil + 1$. Then, we can compute an $\varepsilon$-minimizer in time $\sum_{t=0}^{T-1} \mathrm{Time}(L + 2^{-t}\Delta \mathcal{K}_R^-/R^2, 2^{-t}\Delta \mathcal{K}_R^+/R^2, \mathcal{M}, R)$, where $\mathcal{K}_R^+$ and $\mathcal{K}_R^-$ are constants that depend on $R$ and the bounds on the sectional curvature of $\mathcal{M}$.*

**Example 3.4.** Applying reduction Theorem 3.3 to the algorithm in Corollary 3.2 we can optimize $L$-smooth and g-convex functions defined on $\mathcal{H}$ or $\mathcal{S}$ with a gradient oracle complexity of $\widetilde{O}(L/\sqrt{\varepsilon})$.

Note that this reduction cannot be applied to the locally accelerated algorithm in (Zhang & Sra, 2018), that we discussed in the related work section. The reduction runs in stages by adding decreasing $\mu_i$-strongly convex regularizers until we reach $\mu_i = O(\varepsilon)$. The local assumption required by the algorithm in (Zhang & Sra, 2018) on the closeness to the minimum cannot be guaranteed. In (Ahn & Sra, 2020), the authors give an unconstrained global algorithm whose rates are strictly better than RGD. The reduction could be applied to a constrained version of this algorithm to obtain a method for smooth and g-convex functions defined on manifolds of bounded sectional curvature and whose rates are strictly better than RGD.

## 4 CONCLUSION

In this work we proposed a first-order method with the same rates as AGD, for the optimization of smooth and g-convex or strongly g-convex functions defined on a manifold other than the Euclidean space, up to constants and log factors. We focused on the hyperbolic and spherical spaces, that have constant sectional curvature. The study of geometric properties for the constant sectional curvature case can be usually employed to conclude that a space of bounded sectional curvature satisfies a property that is in between the ones for the cases of constant extremal sectional curvature. Several previous algorithms have been developed for the optimization in Riemannian manifolds of bounded sectional curvature by utilizing this philosophy, for instance Ahn & Sra (2020); Ferreira et al. (2019); Wang et al. (2015); Zhang & Sra (2016; 2018). In future work, we will attempt to use the techniques and insights developed in this work to give an algorithm with the same rates as AGD for manifolds of bounded sectional curvature.

The key technique of our algorithm is the effective lower bound aggregation. Indeed, lower bound aggregation is the main hurdle to obtain accelerated first-order methods defined on Riemannian manifolds. Whereas the process of obtaining effective decreasing upper bounds on the function works similarly as in the Euclidean space—the same approach of locally minimizing the upper bound given by the smoothness assumption is used—obtaining adequate lower bounds proves to be a difficult task. We usually want a simple lower bound such that it, or a regularized version of it, can be easily optimized globally. We also want that the lower bound combines the knowledge that the g-convexity or g-strong convexity provides for all the queried points, commonly an average. These Riemannian convexity assumptions provide simple lower bounds, namely linear or quadratic, but each with respect to each of the tangent spaces of the queried points only. The deformations of the space complicate the aggregation of the lower bounds. Our work deals with this problem by finding appropriate lower bounds via the use of a geodesic map and takes into account the deformations incurred to derive a fully accelerated algorithm. We also needed to deal with other technical problems. Firstly, we

needed a lower bound on the whole function and not only on $F(x^*)$, for which we had to construct two different linear lower bounds, obtaining a relaxation of convexity. Secondly, we had to use an implicit discretization of an accelerated continuous dynamics, since at least the vanilla application of usual approaches like Linear Coupling Allen Zhu & Orecchia (2017) or Nesterov's estimate sequence Nesterov (1983), that can be seen as a forward Euler discretization of the accelerated dynamics combined with a balancing gradient step Diakonikolas & Orecchia (2019), did not work in our constrained case. We interpret that the difficulty arises from trying to keep the gradient step inside the constraints while being able to compensate for a lower bound that is looser by a constant factor.

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
