# OpenReview forum: "Acceleration in Hyperbolic and Spherical Spaces"
_ICLR.cc/2021/Conference — Reject_

### Official Review · AnonReviewer3 · 2020-10-28
**Global rates on constant curvature model spaces via geodesic map analysis and approximate duality gap techniques**

**Rating:** 6
**Confidence:** 2

**Review:**

Summary: This paper provides a generalization of AGD to constant sectional curvature spaces (or subsets of them), and proves the same global rates of convergence that hold in the Euclidean space. Additionally, they provide reductions for the bounded sectional curvature case. Their basic strategy involves the use of geodesic maps to accumulate local linear lower bounds, in a way that accounts for the geometric distortion incurred by the map.

Strengths: The paper is written well and organized in a reasonable fashion. They have a clear description of the general techniques applied in their work, and push overly technical arguments to the appendix. They provide global rates which also apply to g-convex functions (not just strongly convex). Where I have checked, their statements are mathematically sound.

Weaknesses: The domain of applicability for their main rates are restricted to the constant curvature spaces, and it could be argued that it is relatively narrow in scope. I am not sure of the convention in this community, but perhaps it would helpful also to have some experimental results and code to assist in reproduction and discussion of practical import and comparison.

Recommendation: I gave a score of 7, as it seems to provide technical progress over previous results and the authors are clear in describing their contributions. My score is relatively uncertain as I was not able to check many of the technical arguments and lemmas. UPDATE: I would reduce my score to a 6 based on the opinions of my fellow reviewers. It appears that the restricted scope and lack of experimental results is quite a problem within this community and venue.

---

> ### Author Response · Authors · 2020-11-16
> **.**
>
> Thank you for your time and work.

---

### Official Review · AnonReviewer4 · 2020-10-28
**Acceleration in Hyperbolic and Spherical Spaces**

**Rating:** 4
**Confidence:** 2

**Review:**

Unfortunately, though I am familiar with the literature on accelerated gradient methods in Euclidean spaces, I am not familiar enough with Riemannian geometry to provide a confident review of this paper. That being said I can provide some feedback.

Overall, this paper is very dense with mathematics that will not be not particularly familiar to most machine learning people. Moreover the suggested applications to ML (a single line in the paper) is not particularly convincing without more discussion and context (e.g., a fully worked example). This is compounded by the fact that no experiments were presented. This paper would be greatly improved by an experiment that showed the improvement of this algorithm over vanilla gradient approaches on a Riemannian manifold, even if the experiment was totally synthetic. Even better would be to present a real machine learning application. With that in mind, I have a concern that ICLR is not the right venue for this paper. After all, ICLR is focussed on Learning Representations and though I tend to have a relaxed approach to this I think this paper might not be of general enough interest to the ICLR community (even though it may well be an excellent paper).

Some more minor comments:

This paper focuses on hyperbolic and spherical manifolds. How important are these spaces in practical problems? I wonder if the limited scope of the results covers the machine learning applications discussed in the introduction. This needs more discussion. If these spaces are presented primarily because the analysis is easier then this is another reason why ICLR might not be the right venue.

I am surprised that any of the parameters of the manifold do not appear in the bound. Is there some intuition why the bound relies only on the constants of the function f (\mu and L) and not on any property of M (eg, the main results section the results are in terms of L and mu and ignore only factors of log(L/mu))? It would be good to add a discussion of this, or be explicit with the dependency in the bounds if there is one.

Very minor comment: the word 'unfeasible' is unusual (though apparently it is a real world), infeasible is more common.

---

> ### Author Response · Authors · 2020-11-16
> **rebuttal**
>
> The central important assumption that satisfy the applications cited in the introduction (low-rank matrix completion, dictionary learning, optimization with orthogonality constraints, robust covariance estimation, operator scaling and sparse PCA, among others) belong to the bounded sectional curvature case. Also, there are problems that are non-convex but that are g-convex with the appropriate metric, allowing for efficient optimization. Obtaining full global acceleration in the bounded sectional curvature case is a challenging open problem, for which others works, mentioned in the related work, have provided attempts to solve it, resulting in the partial results cited in the paper, published in important Machine Learning conferences ( Liu et al. NeurIPS 2017, Alimisis et al. AISTATS 2020, Zhang and Sra COLT 2018, and Ahn and Sra COLT 2020). These works only provided acceleration when the point is already very close to the minimizer or had rates in between of RGD and AGD. We obtain the first result that achieves full global acceleration in a Riemannian manifold other than the Euclidean space. But not only that, this serves as an important step towards obtaining a solution for the general problem of main interest, as algorithms for the bounded sectional curvature case often rely on the results of the constant curvature case and then compare the properties of the manifolds of bounded sectional curvature \in [K_1, K_2] with the ones of the two manifolds of constant curvature K_1 and K_2, as the 5 papers cited in the first paragraph of the conclusion section.
>
> > Parameters of the manifold.
>
> Note that at the beginning of section 2 we reduced the problem to curvature 1 or -1 and mentioned that the reduction makes the parameters L, mu, and R be rescaled accordingly, and we referenced to Remark C.1. This effectively means that for a manifold of constant sectional curvature K, when reducing the problem to live in a manifold with K = 1 or -1, the new smoothness, strong g-convexity and radius R  are transformed into (L / \sqrt{\abs{K}}, \mu / \sqrt{\abs{K}}, R\sqrt{\abs{K}}), where (L, \mu, R) were the original parameters. And that is where the dependence on the sectional curvature appears.
>
> Thank you for the word suggestion. We hope to have clarified your concerns regarding the context of the problem in the machine learning community and regarding the parameters of the manifold.

---

### Official Review · AnonReviewer2 · 2020-10-29
**This paper obtains the state-of-the-art rates with solid theoretical guarantees on accelerated Riemannian optimization.**

**Rating:** 7
**Confidence:** 1

**Review:**

This paper proposes a global accelerated method on Riemannian manifolds with the same rates as accelerated methods in the Euclidean space up to log factors. Reductions have also been studied on Riemannian manifolds.

Quality: I think this paper has high quality in theory.

Clarity: I have no experience on Riemannian manifolds before. This paper reads difficult for me. I think this paper is too technical and some descriptions are not clear.

Originality: There are a number of works that study the problem of first-order acceleration on Riemannian manifolds. This paper studies the special case of constant sectional curvature, i.e., the hyperbolic and spherical spaces. I am not sure whether there are literatures studying the optimization algorithms (either accelerated or non-accelerated) on the constant sectional curvature before.

Significance: This paper gives the state-of-the-art rates in the special case of constant sectional curvature. I think it is significant.

I have some comments. I have no experience on the optimizaton on Riemannian manifolds before. My comments may be too strict for the analysis on Riemannian manifolds.

1. Previous literatures have studied the optimization on Riemannian manifolds of bounded sectional curvature, while this paper focuses on the special hyperbolic and spherical spaces, that have constant sectional curvature. Is there any literature focusing on the constant sectional curvature before? either accelerated or non-accelerated. What is the critical difference when transforming the analysis on the bounded sectional curvature to constant sectional curvature? Is it a straightforward extension, or very challenging?

2. I am not sure whether each step of the proposed method needs more computations than the standard accelerated gradient method. For example, function f is a composition of F and h^{-1}, can \nabla f(x) be efficiently computed? The BinaryLineSearch needs to compute \Gamma_i^{-1} and x_{i+1}^{\lambda}, do they need more computations?

---

> ### Author Response · Authors · 2020-11-16
> **rebuttal**
>
> > Other works on constant sectional curvature
>
> We are not aware of other works working with the constant sectional curvature case only. The bounded sectional curvature includes the constant sectional curvature so any work applied to the former applies to the latter. Consequently, indeed there are algorithms that do not achieve the global accelerated rates of our work (RGD, and the works mentioned in the related work). But no work has achieved global acceleration in the former setting. There is no accelerated analysis to be transformed from the bounded sectional curvature case. Obtaining global acceleration in that case is an open problem, for which we provide some progress by solving the constant curvature case, which was a highly non-trivial task. We are working on extending our analysis to the bounded curvature case. It is not a straightforward extension, and requires some extra ideas.
>
> > More computations?
>
> Lemma 2.1.c provides a simple way to compute $\nabla F$ using $\nabla f$ and vice versa. So $\nabla f$ can be computed virtually without extra computations, either composing $F$ with $h^{-1}$ so one can use common automatic differentiation engines that are built in ML libraries, or by computing first $\nabla F(x)$ using some Riemannian optimization library like https://www.manopt.org/tutorial.html and then using the formula in Lemma 2.1.c to compute $\nabla f(x)$.
>     About \hat{\Gamma} _ i^{-1}, it has a simple closed formula \hat{\Gamma}_i^{-1}(x) = \frac{a_{i+1}/\gamma_n}{A_i x + a_{i+1}/\gamma_n} coming by (18) in page 19.
>
> The computation of $x_{i+1}^{\lambda}$ requires to compute a gradient of the function $f$. The binary search runs in a logarithmic number of steps, which is the reason why we have a log in the gradient complexity of the algorithm. That is, for instance for g-convexity, the number of iterations of the main for loop to compute an $\epsilon$-minimizer is $O(L/\sqrt{\epsilon})$, and each iteration requires $O(\log(L/\epsilon))$.

---

### Official Review · AnonReviewer1 · 2020-10-30
**Review of Paper699**

**Rating:** 5
**Confidence:** 4

**Review:**

This paper considered the problem of minimizing (strongly and non-strongly) geodesically convex functions on hyperbolic and spherical manifolds, manifolds of constant curvature 1 and -1, respectively, and proposed accelerated algorithms for such problems. In particular, the author(s) showed the proposed algorithms enjoy global accelerated rates that match their Euclidean counterparts. A key to the main result is Lemma 2.2 which asserts a certain quasar convexity-type condition of the pull-back of the objective function to some Euclidean domain through a geodesic map. Based on this lemma, the main result follows from combining techniques for developing accelerated algorithms in Euclidean space, such as the approximate duality gap technique and a certain discretization scheme for continuous dynamics. Some reduction results, which obtain accelerated algorithms for the strongly convex case from the non-strongly convex case, and vice versa, are also presented.

I believe that the technique is new, to the best of my knowledge. And I think that obtaining accelerated algorithms for manifold optimization problems is definitely an important topic. Therefore, the results should be interesting to a broad audience of the conference and deserve some merits. However, I have some doubts about the paper.

1. The presentation of the proofs in the supplementary material is unsatisfactory. There are so many arguments like "trivial/easy to see/prove", "follows straightforwardly/trivially", etc. It makes the proofs very difficult to follow.

2. In Lemma 2.2, it's a bit surprising to me that the constants gamma_p and gamma_n depends only on the radius R of the geodesic ball but not the function F (which implicitly contains K due to the rescaling) nor the point x. Perhaps some intuition of why this is the case and some interpretation of these two constants would be good. It is important as the gamma constants are involved in the complexity in Theorem 2.4 (and hence in Theorem 2.5).

3. The setting of manifolds of constant curvature seems to be quite restrictive. And, as pointed out in the paper, it seems difficult that the technique could be extended to other manifolds.

4. The practicality of the main algorithm (Algorithm 1) seems to be very limited. I understand that this paper focuses more on theoretical side. I am not hoping for practical use either. However, it would be good to demonstrate that performance/behaviour of the proposed algorithm does corroborate with the theories, at least in some toy examples such as optimization problems on the Poincare disk.

Other comments:
1. On page 3, the \tilde{v} (vector of the same norm) is not well-defined. If \tilde{v} satisfies the definition, then it can be checked that 2 \tilde{v} also satisfies the definition.

2. On page 3, it is mentioned that R\ge d(x_0, x^*) implies x^* \in \Exp_{x_0} (\bar{B}(0,R)). I believe such implication requires geodesic completeness of the manifold.

3. On page 3, in the notation section, the constant K appeared without definition.

4. The notions of curvature of manifolds and angles between points on manifolds are used without definition. It would be good to present the definition somewhere in the paper or the supplementary material.

5. A recent paper "An accelerated first-order method for non-convex optimization on manifolds" by Criscitiello and Boumal, which studied accelerated algorithms for non-g-convex optimization on a more general class of manifolds, is missing from the comparison.

6. Sometimes the tangent space T_x M is mistakenly written as T_x. For example, on page 4 and also in the supplementary material.

7. In Lemma 2.3, it is a bit strange that the smoothness property of the pull-back function would require the assumption of the existence of a stationary point. Could you please provide some explanation?

---

> ### Author Response · Authors · 2020-11-16
> **rebuttal R1 part 2/2**
>
> > Smoothness property?
>
> Essentially, if we have a composition f(g(x)) (and let's pick functions from R to R for simplicity) the second derivative is
> f''(g(x)) \cdot g'(x)^2 + f'(g(x)) \cdot g''(x).
> The smoothness constant will be an upper bound on this. Our transformation, depicted by g here, is bounded. f'' is bounded by L by smoothness. But we need to bound f' as well. Since we have a bound R on the distance to the minimizer, and we restrict ourselves to the constrained setting, if there is a point with zero derivative then f' cannot be larger than 2LR. The idea for our case is similar, where we need to be careful to show how those quantities are bounded for our geometric transformation (analogous to bounding g' and g''). So in reality, the condition of stationary point is not needed, but then instead we would need to use the Lipchtiztness constant of the function when restricted to the compact set of points at a distance at most R. We followed the other approach for simplicity, and mentioned this technical detail in the appendix in page 22. In any case, the restriction to the ball of radius R is necessary to bound the deformations of the geometric transformation and that showcases the importance of the accelerated theorem 2.4, which is a result in the Euclidean space, of independent interest, that provides acceleration for the relaxation of convexity (3) in the *constrained case*. As pointed out, it was known how to do this in the unconstrained case (actually for a more general assumption), but new techniques and machinery were necessary to obtain a different algorithm that could achieve acceleration in the constrained case (and only applies to (3)). This is a step forward in the understanding of other conditions that go beyond convexity that can still provide efficient optimization to a minimizer.
>
> Thanks for the typos regarding the late definition of K and regarding T_x M. We have fixed them. Thanks for the feedback on the instances that skip details. We have added and uploaded some extra explanations.

---

> ### Author Response · Authors · 2020-11-16
> **rebuttal R1 part 1/2**
>
>
> > Lemma 2.2. and the constants gamma_p and gamma_n.
>
> The constants that define the relaxation of convexity of the pullback (equation (3)) depend only on the deformations of the manifold, not on the function. Note that R changes with the rescaling too , so the constants depend on R and implicitly on K. What we are doing is, regardless of x, to transform functions y \mapsto a + <v, y-x> defined over T_x M to functions \tilde{y} \mapsto a + <w, \tilde{y} -\tilde{x} > defined in the Euclidean subset B. Here a \in \R, v\in T_x M and the vector w \in R^d satisfies lemma 2.1.c, i.e. v = (1+K\norm{\tilde{x}}^2) w_1 e_1 + \sqrt{1+K\norm{x}^2} w_2 e_2. v does not need to come from a gradient. The deformation of the linear function will depend on the geometry and the only thing that matters is how far x was from the center (hence the dependence on R) and the direction of v (and the constants gamma_n and gamma_p account for the case of the worse direction). Or in other words, the lemma works for any affine function g(y) = a + <v, y-x> by providing another one \tilde{g}(\tilde{y}) = a + <w, \tilde{y} -\tilde{x} > and showing that one bounds the other after multiplication by the constants (which is the result in (2) ). Then g-convexity provides (3) because the gradient of F at x provides one such function g that lower bounds F(x) = f(x). (see last paragraph in last page of the supplementary material). We tried to provide intuition of this in the paragraph below equation (1) in which we mention the lower bound defined by the gradient and how we obtain a looser lower bound that is affine (or linear if we subtract F(x)). We have added a note on this.
>
> >Other related work?
>
> Thank you for the reference, acceleration in non-g-convex functions is a very different problem and thus we cannot compare to the mentioned paper, but we have added to the works mentioned in the introduction.
>
> > Definition of \tilde{v}.
>
> Note that \tilde{v} is defined as having the same norm as v so 2 \tilde{v} does not satisfy the definition.
>
> > R\ge d(x_0, x^*) implies x^* \in \Exp_{x_0} (\bar{B}(0,R)).
>
> We are working with particular underlying manifolds, hyperbolic space and sphere, and they satisfy this property.
>
> > Notions used without definition.
>
> We defined the sectional curvature in the definitions section alluding to the Gauss curvature of a manifold, and provided a reference for an introduction to these concepts. Note we do not use this concept further after establishing we work with hyperbolic and spherical spaces. Note we do not use any angles between points of the manifold. We use angles between vectors in the Euclidean space B to which we map the function and also angles between vectors in T_x M (e.g. Lemma 2.1), which is isomorph to R^d.
>
> > Constant sectional curvature:
>
> The general problem of interest for which several important applications have been found is the setting of bounded sectional curvature. Several papers have studied the problem of acceleration in strongly g-convex functions in this regime and obtained partial results, as shown in the related work. Our result is the first one that obtains global full acceleration for a manifold that is not the Euclidean space and we can also deal with the g-convex case (besides of strongly g-convex). We pointed out our algorithm and ideas are not *straightforwardly* generalizable to bounded sectional curvature, in the sense that the same algorithm does not work, but we are currently working on a generalization of them to apply to the more general case. In any case, this is a first step towards obtaining full acceleration for the more general manifolds with varying but bounded constant curvature and we believe and hope that the new theory developed in this paper will help researchers in the field to find algorithms for more general settings, not necessarily acceleration in g-convex problem in bounded sectional curvature. For instance we also provide Euclidean acceleration when the function satisfies (3) which can model a gradient with multiplicative bounded noise. This work furthers the research on efficient methods for the optimization of some particular non-convex functions and we expect our new theory helps finding more algorithms for efficient non-convex optimization.

---

### Official Review · AnonReviewer5 · 2020-11-06
**Difficult paper to read for AGD on simple geometric models.**

**Rating:** 5
**Confidence:** 2

**Review:**

The paper claims to provide a first order gradient algorithm that achieves (global) equivalent rates of convergence than in Euclidean space on two particular models of geometry (though important) the hypersphere and the hyperbolic space.
The strategy proposed here consists in using the geodesic maps in order to write the minimization problem on the Euclidean space with a controlled distortion, which makes possible the use of a relaxed version of convexity inequalities. In the new coordinate system, the minimised function is not convex but not far from it, in a way the authors are able to control quantitatively.

My opinion on the technical content of the paper is hindered by the difficulty of reading this paper. See the remarks for improving its readability below. The overall result seems a bit weak for all this work (40 pages long paper in total incl. supplementary material.) and the calculations do not seem particularly enlightening.

I would suggest an important rewriting of the paper; Ideally, the main ideas of the paper should be illustrated in the fist technical part section in a simple and enlightening example. Also, putting forward a skeleton of proof of the main result of the paper with more details on the objects would be very helpful for the reader.

Other comments on the readability:

In the contribution section, the third point on reductions is not clear at all. I suggest a rewriting of the sentence that makes it more understandable.

The method uses geodesic maps and in particular  …maps geodesics from the constant curvature space to geodesics in the  Euclidean space… This condition is very stringent. For instance, there is a theorem by Kobayashi which quantifies that affine maps (a condition which implies geodesic maps) are often isometries.
I would suggest the authors spend more time on the definition of a geodesic map and provide a discussion. In particular, they could  write the explicit definition of such maps in the two cases of interest: the (hyper)sphere and the hyperbolic space instead of pointing to the so-called classical geodesic maps.

The beginning of section 2 is difficult to read. The strategy is explained in wordy manner: example:
…. Our approach is to obtain a lower bound that is looser by a constant depending on R, and that is linear over B. In this way the aggregation becomes easier. … Helping the reader with equations or more precise definitions would be helpful.

Section 2.1 is hard to follow, example:
… after applying some desirable modifications, like regularization with a 1-strongly convex function
ψ and removing the unknown x ̃∗ by taking a minimum over X . Note (4) comes from averaging (3) ∗
for y ̃ = x ̃ . …

The contributions to reductions techniques in Section 3 is only accessible to the expert reader and the main text is only statement of the results. The authors could make their point more explicit here.

---

> ### Author Response · Authors · 2020-11-16
> **rebuttal**
>
> > This [using geodesic maps] is very stringent.
>
> This paper proves global acceleration for hyperbolic and spherical spaces and we do not rely on a non-provided existence of some geodesic maps but we provide geodesic maps for our cases and use them. Also note that our maps are not isometries.
>
> > "Our approach is to obtain a lower bound that is looser by a constant depending on R, and that is linear over B. In this way the aggregation becomes easier" help the reader with equations.
>
> Please note we introduce that sentence as intuition with words only to mention again the linear lower bounds before lemmas 2.1 and 2.2 and to provide the equations of the aforementioned lower bounds with equations in (3).
>
> > Section 2.1 is hard to follow. example: "… after applying some desirable modifications".
>
> Section 2.1 is precisely a skeleton of the proof of the main result. It is just a sketch of the proof, where some technical details are skipped to be able to provide the main idea. The full details can be found in the appendix.
>
> > The contributions to reductions techniques are only accessible to the expert reader and the main text is only statement of the results. The authors could make their point more explicit here.
>
> We made explicit the main implications of the contributions by encapsulating them Corollary 3.2 and Example 3.4 which are aimed beyond the expert readers. They say that with a global full accelerated algorithm as our Algorithm 1 (designed for g-convex functions) we can build a full accelerated algorithm for strongly g-convex functions (Corollary 3.2). And with such an algorithm one can obtain a global full algorithm for g-convex functions (Example 3.4). As such, one eliminates the need to design two different ad hoc algorithms and these reductions make possible to focus on any one setting only because the other one will be derived from the former by using the reductions. We added a note saying that the results obtained by corollary 3.2 and example 3.4 provide full global acceleration so they provide this desirable feature.

---

> > ### Comment · AnonReviewer5 · 2020-11-25
> > **Opinion on the authors' rebuttal**
> >
> > I thank the authors for their answers. Their modifications in the paper make it partly more readable. I do not think appropriate to change my rating.

---

### Author Response · Authors · 2020-11-16
**Changes in the manuscript**

Thank you for your time with this review. This message serves as a centralized source for changes in the document with respect to the first upload.

+ We added details of some steps in the proofs that we had ommited, as suggested by Reviewer1.

+ We added the citation Criscitiello and Boumal, 2020, provided by Reviewer1.

+ We added a comment on why the constants gamma_p and gamma_n depend on R only and not on the function, as suggested by Reviewer1.

+ We added a comment to make explicit the fact that the reductions, and in particular the simplified consequences depicted in Corollary 3.2 and Example 3.4, make possible to obtain global full accelerated methods from one set of assumptions from the other without needing any extra effort (where the assumptions are g-convexity and strong g-convexity, along with smoothness), as suggested by Reviewer 5.

+ We rephrased explanation of the reductions appearing in the main results section, as suggested by Reviewer5

+ We corrected a few typos mentioned by the reviewers.

Thank you for your work. We also replied to each reviewer individually. We hope to have clarified the concerns that were raised.

---

### Decision · Program_Chairs · 2021-01-07
**Final Decision**

**Decision:**

Reject

**Comment:**

Reviewers generally appreciate the theoretical contribution of the paper, namely Accelerated Gradient Descent on the sphere and hyperbolic space with the same convergence rate as the Euclidean counterpart. However, there are several major concerns with the current work. From a theoretical standpoint, the geodesic map, which plays a crucial role in the algorithm and theoretical analysis, exists if and only if the manifold has constant sectional curvature (sphere and hyperbolic space). It is not at all clear how the current approach can be extended beyond this setting.
From an algorithmic viewpoint, the stated algorithm has not been experimentally validated. It is suggested that at least some synthetic experiments, e.g. on the sphere or Poincare disk, be carried out. Finally, the current presentation is quite dense and should be considerably improved.